# The Prognosis of Baseline Mitral Regurgitation in Patients with Transcatheter Aortic Valve Implantation

**DOI:** 10.3390/jcm10173974

**Published:** 2021-09-02

**Authors:** Juqian Zhang, Arnaud Bisson, Jad Boumhidi, Julien Herbert, Christophe Saint Etienne, Anne Bernard, Gregory Y.H. Lip, Laurent Fauchier

**Affiliations:** 1Liverpool Centre for Cardiovascular Science, Liverpool Heart and Chest Hospital, University of Liverpool, Liverpool L69 7ZX, UK; juqian.zhang@liverpool.ac.uk (J.Z.); Gregory.Lip@liverpool.ac.uk (G.Y.H.L.); 2Service de Cardiologie, Centre Hospitalier Universitaire et Faculté de Médecine, Université de Tours, CEDEX 1, 37044 Tours, France; arnaud.bisson37@gmail.com (A.B.); Jad.bmd@gmail.com (J.B.); j.herbert@chu-tours.fr (J.H.); csaintetienne@gmail.com (C.S.E.); anne.bernard@univ-tours.fr (A.B.); 3Service de Médecine Interne, Unité d’Endocrinologie Diabétologie et Nutrition, Centre Hospitalier Universitaire et Faculté de Médecine, Université de Tours, CEDEX 1, 37044 Tours, France

**Keywords:** transcatheter aortic valve implantation, mitral regurgitation, tricuspid regurgitation, all-cause mortality, cardiovascular mortality

## Abstract

Mitral regurgitation (MR) is the most common valvular lesion in transcatheter aortic valve implantation (TAVI) recipients. This study aims to assess the long-term prognostic impact of baseline MR in TAVI patients. Methods: Adult patients who underwent TAVI were identified in the French National Hospital Discharge Database. All-cause and cardiovascular mortality, stroke, and rehospitalization with heart failure (HF) were compared in TAVI patients with and without baseline MR and tricuspid regurgitation (TR), respectively; the associations of MR and TR with the outcomes were assessed by Cox regression. Results: Baseline MR was identified in 8240 TAVI patients. Patients with baseline MR have higher yearly incidence of all-cause mortality (HR: 1.192, 95% confidence interval CI: 1.125–1.263), cardiovascular mortality (HR: 1.313, 95%CI: 1.210–1.425), and rehospitalization for heart failure (HF) (HR: 1.411, 95%CI: 1.340–1.486) compared to those without, except for stroke rate (HR: 0.988, 95%CI: 0.868–1.124). Neither baseline MR nor TR was an independent risk predictor for all-cause mortality or cardiovascular mortality in TAVI patients. Baseline MR was independently associated with rehospitalization for HF in TAVI patients. Conclusions: Baseline MR and TR were associated with increased all-cause and cardiovascular mortality post-TAVI, however, neither of them was independent predictor for all-cause or cardiovascular mortality.

## 1. Introduction

Transcatheter aortic valve implantation (TAVI) or replacement (TAVR) is the treatment of choice for symptomatic severe aortic stenosis (AS) in elderly patients with comorbidities and frailty who are not suitable for surgical valve replacement. In this group of patients with high mortality risk, the short-term post-TAVI survival is satisfactory, and the long-term survival is acceptable; in a meta-analysis which included 13,857 TAVI patients with a mean age of 81.5 ± 7.0 years, the aggregated survival was 83%, 48%, and 28% at 1, 5, and 7 years, respectively [1]. Male sex, baseline atrial fibrillation (AF), chronic obstructive pulmonary disease, impaired renal function, diabetes mellitus, coronary artery disease, and heart failure were among the strongest independent risk factors for long-term post-TAVI survival in national registry-based analyses [2,3].

Mitral regurgitation (MR) is the most common valvular lesion in patients who underwent TAVI. In a national registry-based cohort, mild, moderate, and severe mitral regurgitation (MR) at baseline were noted in 47.9%, 31.3% and 5.4% of patients who underwent the procedure, respectively [4]. Of all the TAVI recipients with concomitant MR, functional MR consisted of 36% to 92% of the group [5,6,7,8,9]. Degenerative annular or valvular calcification was the leading cause of organic MR (92%), while 85% of functional MR was secondary to severe aortic stenosis in this group of patients [9]. The reported prevalence of significant (≥moderate) MR varies between 11.5% and 36.8% in cohorts of TAVI patients [4,6,7,10]. Although no significant impact of MR on post-TAVI survival was reported in smaller cohorts [11,12], a meta-analysis demonstrated that significant MR was associated with increased 30-day (pooled odds ratio OR: 1.49, 95% confidence interval CI: 1.16–1.92) and 1-year mortality (pooled hazard ratio HR: 1.32, 95%CI: 1.12–1.55) [13]. Concomitant tricuspid regurgitation (TR), however, is less prevalent than MR in TAVI population, and majority of significant TR are secondary to pulmonary hypertension and right ventricle remodeling [12,14,15,16]. Baseline moderate to severe TR and right ventricular dysfunction are associated with increased all-cause mortality post TAVI [17].

This study was conducted to assess the long-term prognostic impact of baseline MR in patients who underwent TAVI in the French national cohort.

## 2. Method

### 2.1. Patients and Data Collection

A longitudinal national cohort study was conducted based on the National Hospitalization Discharge Database (PMSI, abbreviation for the Programme de Medicalisation des Systemes d’Information in French) which keeps a record of every discharge from public or private hospitals in France. The diagnoses were coded according to the 10th revision of the International Classification of Diseases (ICD-10) in PMSI [18]. Mitral and/or tricuspid regurgitation were presumed to be clinically significant to be mentioned as one of the diagnoses on discharge in PMSI (used for reimbursement), although no clear grading of the regurgitation was available under ICD-10 coding in the database. Anonymized data of all adult patients of ≥18 years of age who had aortic stenosis which was treated with transcatheter TAVI in France between 1 January 2010 and 31 December 2018 were collected following approval by the institutional review board of the Pole Coeur thorax Vaisseaux from the Trousseau University Hospital (Tours, France) on 1 December 2015. The data collection and handling were approved by the independent national ethics committee which protects human rights and freedom in France (Conseil National de L’informatique et des Libertés) under the authorization number 1749007. The study was based on a retrospective collection of anonymized data of patients; therefore, an ethical review was not required, and no impact on their care ensued.

Patients were followed until 31 December 2018 for the occurrence of outcomes. We aimed to evaluate the incidence of all-cause death, cardiovascular death, non-cardiovascular death, all-cause stroke and rehospitalization for heart failure. The endpoints were evaluated with follow-up starting from date of TAVI until date of each specific outcome or date of last news in the absence of the outcome. Information on outcomes during follow-up was obtained by analyzing the PMSI codes for each patient. All-cause death, heart failure, all-cause stroke, and heart failure were identified using their respective ICD-10 or procedure codes. Mode of death (cardiovascular or non-cardiovascular) was identified based on the main diagnosis during hospitalization resulting in death. Rehospitalization was considered to be due to heart failure when heart failure was recorded as the first diagnosis.

### 2.2. Statistical Analysis

Qualitative variables were described using counts and percentages, and continuous quantitative variables were described as mean and standard deviation or median and interquartile range. Comparisons were made using parametric or nonparametric tests, as appropriate: The Wilcoxon signed rank and Kruskal-Wallis tests were used for comparing values between two independent groups, and the χ2 test was used to compare categorical data.

The incidence rates (IR, %/year) with 95% Confidence Interval (95%CI) for each outcome of interest during follow-up was estimated according to the presence of concomitant MR and TR and were compared using hazard ratios (HRs). HRs and asymptotic two-sided 95% confidence intervals (CIs) were estimated using Cox proportional hazards model for each outcome of interest during the whole follow-up. A multivariable analysis was also performed with all baseline characteristics to identify independent characteristics associated with the occurrence of each clinical outcomes.

In all analyses, *p* < 0.05 was considered statistically significant. Analyses were performed using Enterprise Guide 7.1, (SAS Institute Inc., SAS Campus Drive, Cary, NC, USA), USA and STATA version 16.0 (Stata Corp, College Station, TX, USA).

### 2.3. Patient and Public Involvement

No patients were directly involved in developing plans for recruitment, design, or implementation of the study. No patients were asked to advise on interpretation or writing up of results. There were no plans to disseminate the results of the research to study participants (as this was a cohort study), however, results with potential importance may have university/hospital press release, or social media posting for the relevant patient community.

## 3. Results

A total of 42,866 patients (mean age: 83 ± 7 years) with TAVI were included in the cohort, 8240 (19.2%) of these patients have concomitant baseline MR, and 1821 (4.3%) had baseline TR (Figure 1). The mean follow-up for included patients was 1.28 ± 1.58 years (median: 0.63 years, interquartile range: 0.03–2.05). The baseline characteristics of the patients with or without baseline MR or TR are summarized in Table 1 and Appendix A, respectively. TAVI patients with concomitant MR were significantly frailer, with more comorbidities and risk factors for cardiovascular diseases than those with no MR, and similar findings were observed for those with baseline TR compared to those with no TR. During the follow-up post-TAVI, new-onset of MR was reported in discharge diagnosis in 1632 patients (4.2%), meanwhile, 416 patients (0.91%) had new onset of TR.

During the follow-up, the yearly incident all-cause mortality rate was 16.89% (95%CI 16.07–17.75) in TAVI recipients with concomitant MR, as compared to 14.39% (95%CI 14.02–14.78) in those without MR (HR: 1.192, 95%CI: 1.125–1.263) (Appendix A) Likewise, the yearly incidence and risks of cardiovascular mortality and rehospitalization for heart failure were significantly higher in TAVI patients with MR than those without. No difference was observed in the incidence and risk of stroke between the two groups (Appendix A). Cumulative incidences curves with higher risk for all-cause mortality and cardiovascular mortality for TAVI patient with MR compared to those with no baseline MR at baseline are presented in Figure 2. Cumulative incidences curves for ischemic stroke and rehospitalization for HF for TAVI patient with MR compared to those with no baseline MR at baseline are presented in Appendix A.

TAVI patient with concomitant baseline MR and TR had higher all-cause mortality compared to patients with neither of the conditions, and those with isolated MR. However, there was no significant difference in all-cause mortality between patients with concomitant MR/TR and isolated TR (Figure 3). Patients with both MR and TR were noted with the highest cardiovascular mortality as compared to isolated baseline MR or TR and patients with neither of the regurgitations (Figure 3). New-onset MR and new-onset TR were neither predictors of all-cause death (HR: 0.929, 95%CI 0.835–1.032 for new-onset M; HR: 0.910, 95%CI 0.745–1.111 for new-onset TR) or of cardiovascular mortality (HR: 0.863, 95%CI 0.727–1.024 for new-onset MR; HR: 0.863, 95%CI 0.727–1.024 for new-onset TR).

TAVI patient with concomitant baseline MR and TR had similar risk of ischemic stroke compared to patients with neither or one of the conditions (Appendix A). Patients with both MR and TR and those with isolated TR were noted with the highest risk of rehospitalization for HF as compared to those isolated baseline MR, while patients with neither of the regurgitations had the lower risk of being re-hospitalized for HF (Appendix A).

On multivariable Cox regression analysis, neither baseline MR (adjusted HR: 1.010, 95%CI: 0.951–1.072, *p* = 0.75) nor TR (adjusted HR: 1.076, 95%CI: 0.960–1.206, *p* = 0.21) was an independent risk predictor for all-cause mortality in TAVI patients (Table 2). Neither baseline MR (adjusted HR: 1.062, 95%CI: 0.973–1.159, *p* = 0.18) nor TR (adjusted HR: 1.119, 95%CI: 0.953–1.316, *p* = 0.17) was an independent risk predictor for cardiovascular mortality in TAVI patients (Table 3). Baseline MR (adjusted HR: 1.132, 95%CI: 1.071–1.196, *p* < 0.0001), rather than baseline TR (adjusted HR: 1.100, 95%CI: 0.994–1.218, *p* = 0.07), was independently associated with rehospitalization for HF in TAVI patients (Table 4).

## 4. Discussion

The study is one of the largest database-derived cohort studies which assessed the prognostic impact of baseline MR and TR in TAVI recipients in publication. The main findings from the study include: (1) baseline MR was associated with increased all-cause and cardiovascular mortality post-TAVI procedure; however, when adjusted for other baseline characteristics, it was not an independent risk factor for either of the mortality outcomes; (2) baseline TR, like MR, was not an independent risk factor for mortality among TAVI recipients; (3) baseline MR and TR were associated with increased rehospitalization for HF post TAVI, and MR was an independent risk factor for rehospitalization for HF. By contrast, neither of these conditions was associated with a higher risk of ischemic stroke.

The all-cause mortality in severe AS patients increases stepwise with progressive stages of cardiac remodeling: (1) left ventricular dysfunction, (2) significant MR, (3) pulmonary hypertension and significant TR, (4) right heart dilatation and contractile dysfunction [19]. Similar to other national and international registry cohorts, our study noted increased all-cause mortality rates at 1-year and beyond in TAVI patients with preprocedural significant MR, as compared to those with lesser degrees or no MR [4,7,20,21,22,23]. However, conflicting results were found on multivariable analysis with some studies reporting significant MR as an independent predictor for all-cause mortality at 1-year and beyond [7,20,21], while others including this study did not [4,22,23]. Despite the clinical risk profile of patients with significant MR which predisposes them to death and cardiovascular events compared with those with mild or no MR, the majority of MR may improve following the TAVI procedure. Swedish national registry-based study noted that patients whose significant baseline MR improved following the TAVI had no increased 5-year mortality compared with mild or no MR at baseline; whereas patients whose significant MR remained unchanged or worsened post-procedure had approximately a 1.7-fold and 2-fold increase in 5-year mortality, respectively [20]. This suggests that preprocedural MR has diverse prognostic potentials depending on the presence of other clinical risk factors for TAVI patients, it is possibly a surrogate marker of other comorbidities than an independent risk predictor. Patients with “true” high risk for mortality are those patients with post-procedural, rather than pre-procedural significant MR [24]. It is worth noting that neither short-term nor long-term mortality risk models commonly used for TAVI patients incorporated baseline MR into their covariates [25,26]. Due to the observations of a significant proportion of baseline MR regressing following TAVI procedure, as well as the lack of convincing evidence over the prognostic benefit of simultaneous TAVI and mitral valve clipping for this group of patients with multiple comorbidities and marked risk of adverse events, the intervention on significant MR alongside TAVI is not routinely recommended [27]. A sequential treatment of significant MR following successful TAVI could be potentially beneficial for those who remains symptomatic following the procedure [28]. An ongoing clinical trial (MITAVI, NCT04009434) which aims to assess the efficacy of additional mitral valve clipping in TAVI patients with significant MR could potentially fill in the gap of evidence [29].

Less prevalent than MR, significant (≥moderate) TR affects 11–27% of TAVI recipients in published cohorts [12,14,15]. Although patients with significant TR have a 1.5 to 2-fold increase in long-term mortality, whether significant TR is an independent risk factor for all-cause mortality at or beyond 1-year is a topic of debate [12,14,15]. Prior studies suggested an interaction between the severity of MR and TR in TAVI patients, and significant TR was associated with 6-month all-cause mortality only in those with non-significant MR [14,30]. In the current study, presumably significant TR does not carry an independent risk for long-term postprocedural all-cause mortality, which adds further evidence to the topic. Cardiovascular causes account for approximately 50% and 60% of all-cause mortality in TAVI patients at 6-month [31] and 4-year [32] post-procedure, respectively. However, limited studies so far have assessed the association between baseline TR and cardiovascular mortality in multivariate analysis in TAVI recipients. The current study could potentially provide some evidence in the less-studied area.

The prognosis of TAVI procedures varies with different valves and surgical/interventional approaches: the SAPIEN 3 valves may be associated with lower mortality and all-cause hospitalization compared with Evolut R and CoreValve [18,33]; higher incidence of stroke and 12-month mortality was noted in trans-apical and direct-aortic procedures as opposed to surgical-femoral and subclavian approach [34]. Although no direct comparison in the clinical outcomes was performed in-between different prosthetic valves in current study, however, the Sapien 3 valve was independently associated with consistently reduced all-cause mortality, cardiovascular mortality, and re-hospitalization for HF in multivariate analyses, whereas other TAVI valves were not independent risk predictors for cardiovascular mortality (Table 2, Table 3 and Table 4). One of the strengths of the current study is that it reflects the real-world data of TAVI patients who have undergone a mixture of prosthetic valves via a percutaneous approach on a nationwide scale. A few risk-predictive models for TAVI patients have been developed to assess the patients for their suitability for the procedure, however, none of them were able to effectively differentiate patients for whom the procedure could be futile, partly due to considerate uncertainty in the outcomes of patients with high risk [35]. Future studies in developing risk models for TAVI which assesses and incorporates pre-procedural MR/TR alongside other risk factors in could potentially provide an effective risk stratification for TAVI candidates.

### Limitations

The current study has several limitations. One major limitation is the uncertainty in the severity of mitral and tricuspid regurgitation through the coded diagnosis in the database. An accurate assessment and grading of the regurgitation in those with baseline mitral and tricuspid regurgitation were unavailable in current study. Increased severity of baseline mitral and tricuspid regurgitation is associated with stepwise increase in short-term comorbidities and worse long-term survival, as well as more rehospitalization for HF in TAVI recipients [36,37]. The lack of information in the specific grading of atrioventricular regurgitation limits the potential extrapolation of the findings from current study into pre-operative risk assessment of TAVI candidates, given the significant different prognostic profile between those with insignificant (none or mild) and significant (moderate or severe) MR or TR [36,37]. Meanwhile, the regression of significant baseline MR and TR was observed in approximately 50–60% of TAVI patients following the procedure, and the post-procedurally regression in regurgitation was significantly associated with better clinical outcomes [36,38]. The unavailability of the dynamic changes of atrioventricular regurgitation over the time course in our study limits further in-depth analysis of the risk factors contributing to prognosis post TAVI. Besides, the coded diagnosis from database could potentially underestimate the prevalence of mitral and tricuspid regurgitations as well as the incidence of outcomes reported in the cohort. Future cohort studies with accurate assessment and monitor of baseline and post-procedural significant MR and TR could be helpful in the evaluation of their prognostic implications based on the explorative findings from the current study. Furthermore, there is a limitation in extrapolating the findings from this cohort into other populations with different ethnicities, socioeconomic status, and level of service beyond this healthcare system, as well as the epidemiology of TAVI recipients and available TAVI procedures. With introduction of new TAVI valves and improvement in implantation skills, remarkably improved survival and reduced complications were noted in TAVI recipients [39]. The current study included different generations of prosthetic valves over an extended period, and no comparison between the first- and second-generation valves, neither did we perform a comparison prior and post the introduction of the second-generation valves in 2015/2016. Caution is required in extrapolating the observations from current study in institutions and countries where different prostheses and procedures are adopted.

## 5. Conclusions

Baseline MR was associated with increased all-cause mortality, cardiovascular mortality, and rehospitalization post-TAVI, but was not associated with a higher rate of stroke. Neither baseline MR nor TR was an independent predictor for all-cause mortality or cardiovascular mortality. Baseline MR was an independent predictor for rehospitalization for HF post-TAVI.

## Figures and Tables

**Figure 1 jcm-10-03974-f001:**
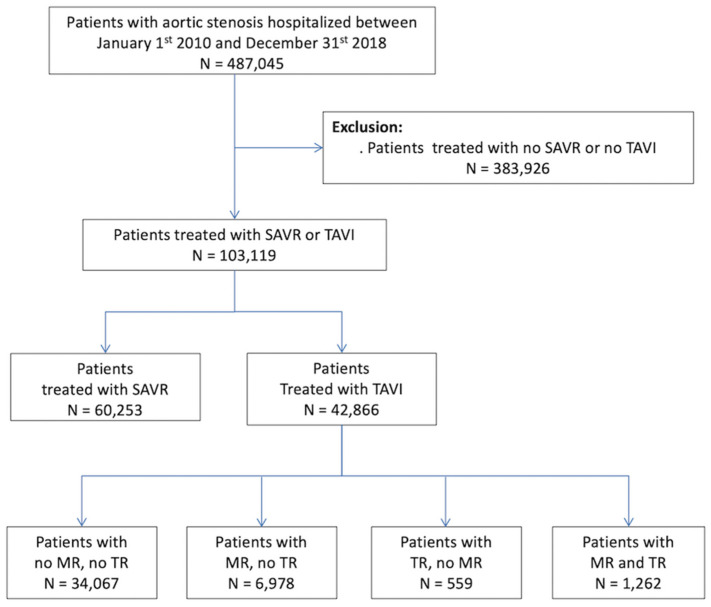
Flow chart of the study patients. MR: mitral regurgitation; SAVR: surgical aortic valve replacement; TAVI: transcatheter aortic valve implantation; TR: tricuspid regurgitation.

**Figure 2 jcm-10-03974-f002:**
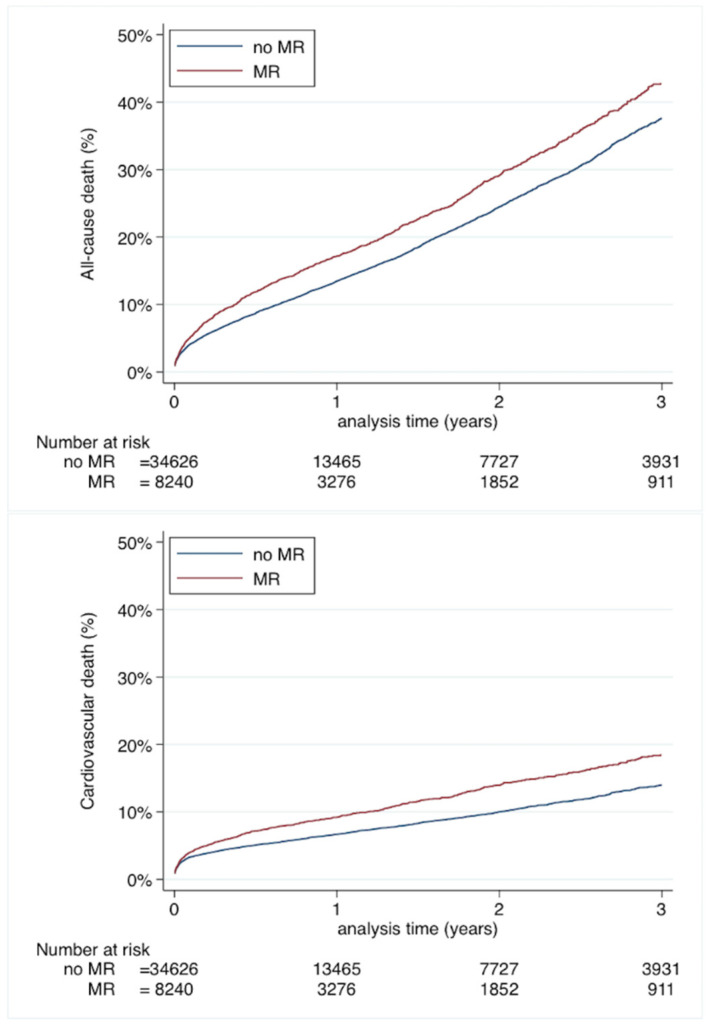
Cumulative incidence for all-cause mortality (top panel) and cardiovascular mortality (lower panel) of patients treated with TAVI with MR or no MR at baseline.

**Figure 3 jcm-10-03974-f003:**
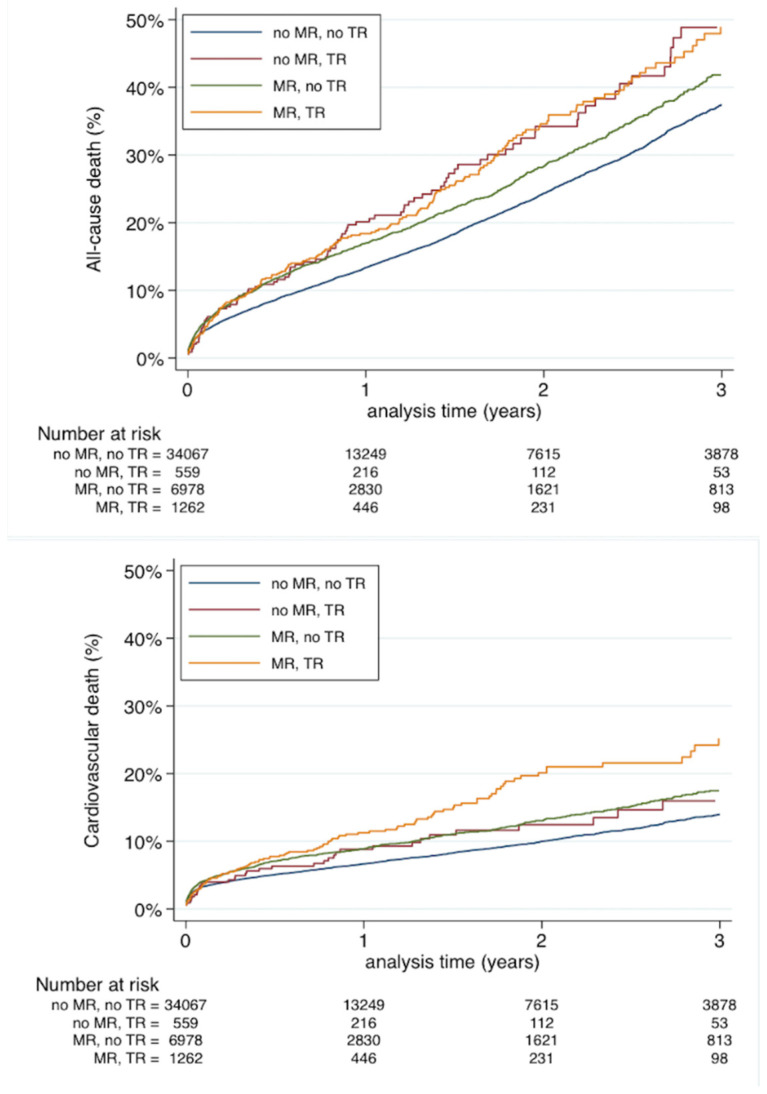
Cumulative incidences for all-cause mortality (top panel) and cardiovascular death (lower panel) of patients treated with TAVI according to MR (or no MR) and TR (or no TR) at baseline.

**Table 1 jcm-10-03974-t001:** Baseline characteristics of patients treated with TAVI according to baseline mitral regurgitation.

	No Mitral Regurgitation	Mitral Regurgitation	*p*	Total
	(*n* = 34,626)	(*n* = 8240)		(*n* = 42,866)
Age, years	82.7 ± 6.7	82.7 ± 7.0	0.5	82.7 ± 6.8
Sex (female)	17,587 (50.8)	4382 (53.2)	0.0001	21,969 (51.3)
Charlson comorbidity index	3.9 ± 2.8	4.6 ± 2.8	<0.0001	4.0 ± 2.8
Frailty index	5.2 ± 5.7	6.4 ± 6.0	<0.0001	5.4 ± 5.8
EuroSCORE II	3.6 ± 0.9	4.0 ± 1.0	<0.0001	3.7 ± 1.0
Hypertension	27,618 (79.8)	6918 (84.0)	<0.0001	34,536 (80.6)
Diabetes mellitus	10,157 (29.3)	2439 (29.6)	0.63	12,596 (29.4)
Heart failure with congestion	18,312 (52.9)	5871 (71.3)	<0.0001	24,183 (56.4)
History of pulmonary oedema	1591 (4.6)	658 (8.0)	<0.0001	2249 (5.2)
Aortic regurgitation	3162 (9.1)	1975 (24.0)	<0.0001	5137 (12.0)
Mitral regurgitation	0 (0.0)	8240 (100.0)	-	8240 (19.2)
Previous endocarditis	189 (0.5)	108 (1.3)	<0.0001	297 (0.7)
Dilated cardiomyopathy	4839 (14.0)	2064 (25.0)	<0.0001	6903 (16.1)
Coronary artery disease	20,760 (60.0)	5593 (67.9)	<0.0001	26,353 (61.5)
Previous myocardial infarction	4618 (13.3)	1460 (17.7)	<0.0001	6078 (14.2)
Previous PCI	9834 (28.4)	2599 (31.5)	<0.0001	12,433 (29.0)
Previous CABG	2788 (8.1)	839 (10.2)	<0.0001	3627 (8.5)
Vascular disease	12,001 (34.7)	3653 (44.3)	<0.0001	15,654 (36.5)
Atrial fibrillation	14,684 (42.4)	4691 (56.9)	<0.0001	19,375 (45.2)
Previous pacemaker or ICD	6744 (19.5)	2085 (25.3)	<0.0001	8829 (20.6)
Ischemic stroke	1837 (5.3)	492 (6.0)	0.02	2329 (5.4)
Intracranial bleeding	482 (1.4)	154 (1.9)	0.001	636 (1.5)
Smoker	2512 (7.3)	836 (10.1)	<0.0001	3348 (7.8)
Dyslipidemia	15,793 (45.6)	4237 (51.4)	<0.0001	20,030 (46.7)
Obesity	8638 (24.9)	2257 (27.4)	<0.0001	10,895 (25.4)
Alcohol related diagnoses	1439 (4.2)	380 (4.6)	0.07	1819 (4.2)
Abnormal renal function	5482 (15.8)	2004 (24.3)	<0.0001	7486 (17.5)
Lung disease	7888 (22.8)	2274 (27.6)	<0.0001	10,162 (23.7)
Sleep apnea syndrome	3030 (8.8)	778 (9.4)	0.05	3808 (8.9)
COPD	5002 (14.4)	1423 (17.3)	<0.0001	6425 (15.0)
Liver disease	1552 (4.5)	556 (6.7)	<0.0001	2108 (4.9)
Gastroesophageal reflux	1126 (3.3)	333 (4.0)	0.0004	1459 (3.4)
Thyroid diseases	4487 (13.0)	1469 (17.8)	<0.0001	5956 (13.9)
Inflammatory disease	3268 (9.4)	1091 (13.2)	<0.0001	4359 (10.2)
Anemia	9006 (26.0)	2822 (34.2)	<0.0001	11,828 (27.6)
Previous cancer	6454 (18.6)	1558 (18.9)	0.57	8012 (18.7)
Edwards Sapien XT^TM^	3425 (9.9)	807 (9.8)	0.79	4232 (9.9)
Edwards Sapien 3	17,552 (50.7)	4090 (49.6)	0.09	21,642 (50.5)
Medtronic Corevalve	4229 (12.2)	1046 (12.7)	0.23	5275 (12.3)
Medtronic Evolut	9420 (27.2)	2297 (27.9)	0.22	11,717 (27.3)
Self-expandable TAVI	13,649 (39.4)	3343 (40.6)	0.05	16,992 (39.6)
Balloon-expandable TAVI	20,977 (60.6)	4897 (59.4)	0.05	25,874 (60.4)

Values are *n* (%) or mean ± SD. CABG: coronary artery bypass graft; COPD: chronic obstructive pulmonary disease; ICD: Implantable Cardioverter Defibrillator; PCI: percutaneous coronary intervention; TAVI: transcatheter aortic valve implantation.

**Table 2 jcm-10-03974-t002:** Predictors of all-cause death in patients treated with TAVI.

	Univariate Analysis		Multivariable Analysis	
	HR, 95%CI	*p*	HR, 95%CI	*p*
Age, years	1.005 (1.002–1.009)	0.002	1.016 (1.012–1.020)	<0.0001
Charlson comorbidity index	1.110 (1.101–1.118)	<0.0001	1.090 (1.077–1.103)	<0.0001
Frailty index	1.165 (1.136–1.195)	<0.0001	0.936 (0.907–0.965)	<0.0001
Sex (male)	1.218 (1.163–1.277)	<0.0001	1.146 (1.088–1.207)	<0.0001
Hypertension	1.046 (0.972–1.126)	0.23	0.903 (0.836–0.976)	0.01
Diabetes mellitus	1.115 (1.062–1.170)	<0.0001	0.906 (0.856–0.959)	0.001
Heart failure with congestion	1.761 (1.671–1.855)	<0.0001	1.319 (1.246–1.398)	<0.0001
History of pulmonary edema	2.544 (2.365–2.735)	<0.0001	1.948 (1.805–2.102)	<0.0001
Aortic regurgitation	1.081 (1.010–1.156)	0.02	0.967 (0.902–1.036)	0.34
Mitral regurgitation	1.181 (1.116–1.249)	<0.0001	1.010 (0.951–1.072)	0.75
Tricuspid regurgitation	1.272 (1.140–1.419)	<0.0001	1.076 (0.960–1.206)	0.21
Coronary artery disease	1.092 (1.038–1.149)	0.001	0.933 (0.878–0.990)	0.02
Previous myocardial infarction	1.288 (1.211–1.369)	<0.0001	1.009 (0.935–1.088)	0.82
Previous PCI	1.133 (1.077–1.192)	<0.0001	1.065 (1.005–1.128)	0.03
Previous CABG	0.948 (0.876–1.027)	0.19	0.953 (0.876–1.037)	0.26
Vascular disease	1.296 (1.236–1.359)	<0.0001	1.075 (1.014–1.140)	0.02
Atrial fibrillation	1.585 (1.511–1.661)	<0.0001	1.341 (1.276–1.409)	<0.0001
Previous pacemaker or ICD	1.182 (1.119–1.248)	<0.0001	0.997 (0.943–1.054)	0.92
Ischemic stroke	1.255 (1.138–1.384)	<0.0001	1.050 (0.950–1.160)	0.34
Smoker	1.194 (1.107–1.287)	<0.0001	1.013 (0.935–1.098)	0.75
Dyslipidemia	0.917 (0.875–0.961)	<0.0001	0.857 (0.815–0.902)	<0.0001
Obesity	0.989 (0.940–1.041)	0.68	0.919 (0.869–0.972)	0.003
Alcohol related diagnoses	1.200 (1.095–1.316)	<0.0001	0.941 (0.849–1.042)	0.24
Abnormal renal function	1.471 (1.393–1.553)	<0.0001	1.054 (0.994–1.117)	0.08
Lung disease	1.408 (1.339–1.479)	<0.0001	1.173 (1.112–1.238)	<0.0001
Sleep apnea syndrome	1.242 (1.151–1.340)	<0.0001	1.099 (1.012–1.192)	0.03
Liver disease	1.583 (1.452–1.726)	<0.0001	1.164 (1.056–1.283)	0.002
Thyroid diseases	1.098 (1.028–1.173)	0.005	0.969 (0.905–1.037)	0.36
Inflammatory disease	1.295 (1.207–1.390)	<0.0001	1.131 (1.052–1.216)	0.001
Anemia	1.458 (1.389–1.531)	<0.0001	1.176 (1.116–1.238)	<0.0001
Previous cancer	1.375 (1.301–1.453)	<0.0001	1.059 (0.995–1.126)	0.07
Edwards Sapien XT	0.869 (0.817–0.923)	<0.0001	0.783 (0.721–0.850)	<0.0001
Edwards Sapien 3	0.998 (0.949–1.049)	0.93	0.875 (0.817–0.937)	<0.0001
Medtronic Corevalve	1.039 (0.982–1.099)	0.18	0.839 (0.776–0.907)	<0.0001
Medtronic Evolut	1.124 (1.055–1.198)	<0.0001	1.000	

CABG: coronary artery bypass graft; COPD: chronic obstructive pulmonary disease; ICD: Implantable Cardioverter Defibrillator; PCI: percutaneous coronary intervention.

**Table 3 jcm-10-03974-t003:** Predictors of cardiovascular mortality in patients treated with TAVI.

	Univariate Analysis		Multivariable Analysis	
	HR, 95%CI	*p*	HR, 95%CI	*p*
Age, years	1.011 (1.005–1.016)	<0.0001	1.020 (1.014–1.026)	<0.0001
Charlson comorbidity index	1.069 (1.057–1.082)	<0.0001	1.014 (0.995–1.034)	0.16
Frailty index	1.088 (1.048–1.130)	<0.0001	0.862 (0.822–0.903)	<0.0001
Sex (male)	1.041 (0.971–1.116)	0.26	0.955 (0.884–1.032)	0.25
Hypertension	0.982 (0.884–1.090)	0.73	0.835 (0.747–0.933)	0.001
Diabetes mellitus	1.101 (1.023–1.184)	0.01	1.032 (0.947–1.124)	0.48
Heart failure with congestion	2.142 (1.975–2.323)	<0.0001	1.661 (1.517–1.819)	<0.0001
History of pulmonary edema	3.631 (3.301–3.994)	<0.0001	2.732 (2.470–3.021)	<0.0001
Aortic regurgitation	1.150 (1.042–1.270)	0.006	0.982 (0.887–1.088)	0.73
Mitral regurgitation	1.313 (1.210–1.425)	<0.0001	1.062 (0.973–1.159)	0.18
Tricuspid regurgitation	1.396 (1.196–1.630)	<0.0001	1.119 (0.953–1.316)	0.17
Coronary artery disease	1.134 (1.051–1.224)	0.001	0.918 (0.838–1.005)	0.06
Previous myocardial infarction	1.473 (1.350–1.608)	<0.0001	1.055 (0.946–1.176)	0.34
Previous PCI	1.208 (1.121–1.302)	<0.0001	1.110 (1.018–1.210)	0.02
Previous CABG	1.120 (1.000–1.254)	0.05	1.098 (0.973–1.240)	0.13
Vascular disease	1.439 (1.342–1.544)	<0.0001	1.255 (1.149–1.369)	<0.0001
Atrial fibrillation	1.624 (1.513–1.743)	<0.0001	1.332 (1.237–1.435)	<0.0001
Previous pacemaker or ICD	1.219 (1.124–1.322)	<0.0001	1.007 (0.927–1.094)	0.87
Ischemic stroke	1.440 (1.257–1.648)	<0.0001	1.372 (1.193–1.576)	<0.0001
Smoker	1.070 (0.952–1.204)	0.26	0.970 (0.856–1.099)	0.63
Dyslipidemia	0.975 (0.909–1.046)	0.48	0.919 (0.851–0.991)	0.03
Obesity	0.948 (0.878–1.024)	0.18	0.892 (0.818–0.971)	0.009
Alcohol related diagnoses	0.910 (0.779–1.064)	0.24	0.799 (0.674–0.948)	0.01
Abnormal renal function	1.583 (1.462–1.714)	<0.0001	1.212 (1.110–1.323)	<0.0001
Lung disease	1.413 (1.312–1.523)	<0.0001	1.266 (1.168–1.373)	<0.0001
Sleep apnea syndrome	1.290 (1.154–1.442)	<0.0001	1.220 (1.081–1.377)	0.001
Liver disease	1.532 (1.343–1.747)	<0.0001	1.390 (1.203–1.608)	<0.0001
Thyroid diseases	1.099 (0.996–1.213)	0.06	0.925 (0.835–1.024)	0.13
Inflammatory disease	1.277 (1.149–1.419)	<0.0001	1.181 (1.060–1.317)	0.003
Anemia	1.382 (1.285–1.487)	<0.0001	1.125 (1.041–1.216)	0.003
Previous cancer	1.072 (0.981–1.171)	0.12	0.990 (0.897–1.093)	0.85
Edwards Sapien XT	0.981 (0.893–1.077)	0.68	0.983 (0.872–1.109)	0.79
Edwards Sapien 3	0.843 (0.783–0.907)	<0.0001	0.875 (0.793–0.966)	0.008
Medtronic Corevalve	1.225 (1.126–1.332)	<0.0001	1.067 (0.953–1.195)	0.26
Medtronic Evolut	1.057 (0.966–1.158)	0.23	1.000	-

CABG: coronary artery bypass graft; COPD: chronic obstructive pulmonary disease; ICD: Implantable Cardioverter Defibrillator; PCI: percutaneous coronary intervention.

**Table 4 jcm-10-03974-t004:** Predictors of rehospitalization for HF in patients treated with TAVI.

	Univariate Analysis		Multivariable Analysis	
	HR, 95%CI	*p*	HR, 95%CI	*p*
Age, years	1.014 (1.010–1.017)	<0.0001	1.021 (1.017–1.025)	<0.0001
Charlson comorbidity index	1.109 (1.101–1.117)	<0.0001	1.047 (1.035–1.059)	<0.0001
Frailty index	1.468 (1.433–1.503)	<0.0001	1.185 (1.151–1.220)	<0.0001
Sex (male)	0.982 (0.939–1.027)	0.42	0.961 (0.915–1.010)	0.12
Hypertension	1.755 (1.613–1.909)	<0.0001	1.283 (1.175–1.400)	<0.0001
Diabetes mellitus	1.381 (1.319–1.445)	<0.0001	1.109 (1.052–1.170)	<0.0001
Heart failure with congestion	2.614 (2.479–2.757)	<0.0001	1.828 (1.726–1.936)	<0.0001
History of pulmonary edema	1.631 (1.496–1.778)	<0.0001	1.075 (0.984–1.174)	0.11
Aortic regurgitation	1.147 (1.076–1.222)	<0.0001	0.975 (0.913–1.040)	0.44
Mitral regurgitation	1.411 (1.340–1.486)	<0.0001	1.132 (1.071–1.196)	<0.0001
Tricuspid regurgitation	1.509 (1.369–1.663)	<0.0001	1.100 (0.994–1.218)	0.07
Coronary artery disease	1.334 (1.269–1.403)	<0.0001	1.143 (1.079–1.211)	<0.0001
Previous myocardial infarction	1.385 (1.308–1.468)	<0.0001	1.095 (1.019–1.177)	0.01
Previous PCI	1.105 (1.052–1.160)	<0.0001	0.963 (0.912–1.017)	0.18
Previous CABG	1.149 (1.070–1.233)	<0.0001	1.126 (1.043–1.215)	0.002
Vascular disease	1.241 (1.187–1.299)	<0.0001	0.913 (0.862–0.966)	0.002
Atrial fibrillation	2.078 (1.985–2.176)	<0.0001	1.630 (1.554–1.709)	<0.0001
Previous pacemaker or ICD	1.482 (1.410–1.558)	<0.0001	1.197 (1.138–1.260)	<0.0001
Ischemic stroke	1.126 (1.022–1.241)	0.02	0.862 (0.781–0.952)	0.003
Smoker	1.122 (1.042–1.208)	0.002	1.014 (0.937–1.097)	0.73
Dyslipidemia	1.150 (1.100–1.203)	<0.0001	1.003 (0.956–1.052)	0.90
Obesity	1.326 (1.265–1.389)	<0.0001	1.112 (1.056–1.171)	<0.0001
Alcohol related diagnoses	1.080 (0.985–1.184)	0.10	0.990 (0.895–1.095)	0.84
Abnormal renal function	1.544 (1.466–1.626)	<0.0001	1.063 (1.006–1.124)	0.03
Lung disease	1.393 (1.327–1.461)	<0.0001	1.149 (1.091–1.210)	<0.0001
Sleep apnea syndrome	1.343 (1.251–1.441)	<0.0001	1.048 (0.971–1.131)	0.23
Liver disease	1.176 (1.069–1.293)	0.001	0.938 (0.846–1.041)	0.23
Thyroid diseases	1.373 (1.295–1.457)	<0.0001	1.062 (0.999–1.129)	0.05
Inflammatory disease	1.256 (1.173–1.344)	<0.0001	0.994 (0.928–1.066)	0.88
Anemia	1.302 (1.242–1.365)	<0.0001	0.992 (0.944–1.043)	0.76
Previous cancer	0.957 (0.903–1.014)	0.14	0.864 (0.810–0.920)	<0.0001
Edwards Sapien XT	0.799 (0.752–0.850)	<0.0001	0.609 (0.564–0.658)	<0.0001
Edwards Sapien 3	0.950 (0.907–0.995)	0.03	0.773 (0.728–0.822)	<0.0001
Medtronic Corevalve	1.008 (0.954–1.065)	0.78	0.702 (0.654–0.755)	<0.0001
Medtronic Evolut	1.331 (1.259–1.408)	<0.0001	1.000	-

CABG: coronary artery bypass graft; COPD: chronic obstructive pulmonary disease; ICD: Implantable Cardioverter Defibrillator; PCI: percutaneous coronary intervention.

## Data Availability

Data may be obtained from a third party and are not publicly available. The data and study materials will not be made available to other researchers for purposes of reproducing the results or replicating the procedure. As this study used data from human subjects, the data and everything pertaining to the data are governed by the French Health Agencies and cannot be made available to other researchers.

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
