# Peer review of "The Prognosis of Baseline Mitral Regurgitation in Patients with Transcatheter Aortic Valve Implantation"

_jcm, 2021, doi:10.3390/jcm10173974_

Round 1
Reviewer 1 Report
The manuscript is well written and of importance for the patients with multivalvular disease. However, I do have some comments.
- In the introduction you highlight mitralvalve regurgitation but don't say anything about tricuspidvalve regurgitation, while in the results and discussion you do report the findings for TR. I would suggest add a little information about tricuspidvalve regurgitation in the introduction to be consistent and use it in your research question.
- It is also a limitation that all TAVI procedures were included from 2010 untill 2018. From 2014/2015 newer valves have dominated the market and did improve survival. Did you look at outcome for the different time periods? Otherwise add it to the limitations.
- For the figures add a label to the y-axis.
- Use the same steps and cut-off values on the y-axis so comparing them is easier. Especially for figure 2 and 3, but also supplementary figure 1.
Author Response
Reviewer 1: The manuscript is well written and of importance for the patients with multivalvular disease. However, I do have some comments.
- In the introduction you highlight mitralvalve regurgitation but don't say anything about tricuspidvalve regurgitation, while in the results and discussion you do report the findings for TR. I would suggest add a little information about tricuspidvalve regurgitation in the introduction to be consistent and use it in your research question.
Response: Many thanks for your suggestion. We have added the following content to the introduction section:
Concomitant tricuspid regurgitation (TR), however, is less prevalent than MR in TAVI population, and majority of significant TR are secondary to pulmonary hypertension and right ventricle remodeling(Hutter 2013;Lindman 2015; Schwartz 2017; Khan 2020). Baseline moderate to severe TR and right ventricular dysfunction are associated with increased all-cause mortality post TAVI (Fan 2019).
2. It is also a limitation that all TAVI procedures were included from 2010 untill 2018. From 2014/2015 newer valves have dominated the market and did improve survival. Did you look at outcome for the different time periods? Otherwise add it to the limitations.
Response: Many thanks for your comment. We haven’t done the subgroup analysis based on different period for the study. We felt that apart from different valves adopted at different time period, there are also other factors such as a shift from very high risk to intermediate-risk candidates for TAVI which could potentially bias the time-period analysis, also this could add complexity for the interpretation of our results. Following your suggestion, we have added the following content to the Discussion section:
Although no direct comparison in the clinical outcomes was performed in-between different prosthetic valves in current study, however, the Sapien 3 valve was independently associated with consistently reduced all-cause mortality, cardiovascular mortality, and rehospitalization for HF in multivariate analyses, whereas other TAVI valves were not independent risk predictors for cardiovascular mortality (Table 2, 3 and 4).
We also added the following to the Limitations in the Discussion section.
With introduction of new TAVI valves and improvement in implantation skills, remarkably improved survival and reduced complications were noted in TAVI recipients (Winter 2020). The current study included different generations of prosthetic valves over an extended period, and no comparison between the first- and second-generation valves, neither did we perform a comparison prior and post the introduction of the second-generation valves in 2015/2016. Caution is required in extrapolating the observations from current study in institutions and countries where different prostheses and procedures are adopted.
3. For the figures add a label to the y-axis.
Response: Thank you for your suggestion. We have added the labels to the y-axis in figures.
4. Use the same steps and cut-off values on the y-axis so comparing them is easier. Especially for figure 2 and 3, but also supplementary figure 1.
Response: Many thanks for your suggestions. We have amended the figures accordingly.
Reviewer 2 Report
In the submitted manuscript, Zhang et al. and his Co-authors present data from a large nationalwide database of French which includes al TAVR-patients with concomitant MR/TR. Adjusted analysis showed that the presence of baseline MR and TR had a signifcant impact on patient´s symptoms but were not independent predictors of all-cause mortality.
The authors are to be congratulated on their manuscript, yet their are several aspects, which need to be adressed before the results can be properly interpreted:
Major comments:
- It is indeed a major limitation, that the severity of regurgitation cannot be provided, as relevant MR and TR has been shown to have both independently an effect on the outcome of TAVR patients. Please discuss this issue in more detail in the discussion section.
Mauri V, Körber MI, Kuhn E, Schmidt T, Frerker C, Wahlers T, Rudolph TK, Baldus S, Adam M, Ten Freyhaus H. Prognosis of persistent mitral regurgitation in patients undergoing transcatheter aortic valve replacement. Clin Res Cardiol. 2020 Oct;109(10):1261-1270. doi: 10.1007/s00392-020-01618-9. Epub 2020 Feb 18. PMID: 32072263; PMCID: PMC7515951.
McCarthy, Fenton H.; Vemulapalli, Sreekanth; Li, Zhuokai; Thourani, Vinod; Matsouaka, Roland A.; Desai, Nimesh D.; Kirtane, Ajay; Anwaruddin, Saif; Williams, Matthew L.; Giri, Jay; Vallabhajosyula, Prashanth; Li, Robert H.; Herrmann, Howard C.; Bavaria, Joseph E.; Szeto, Wilson Y. (2018). The Association of Tricuspid Regurgitation With Transcatheter Aortic Valve Replacement Outcomes: A Report From The Society of Thoracic Surgeons/American College of Cardiology Transcatheter Valve Therapy Registry. The Annals of Thoracic Surgery, (), S0003497517315187–. doi:10.1016/j.athoracsur.2017.11.018
- Page 11: "Our results also suggest that physicians in charge of patients with symptomatic AS and MR at a nationwide level properly select the indication for TAVI (instead of possibly an aortic valve surgery with an associated intervention on the mitral valve) since the mortality outcomes after TAVI in the multivariable analysis were not significantly affected by a baseline MR or TR."
This statement is a bit too strong as most of the patients were not sent to surgery because of the high perioperative risk, as you mentioned in the introduction. Current studies (MITAVI; NCT04009434) are investigating whether patients who have had successful transfemoral transcatheter aortic valve implantation (TAVI) with simultaneous moderate to severe mitral valve regurgitation (MR) benefit from additional treatment for mitral valve regurgitation. Please add and discuss this point in the discussion section.
- Conclusion section: The authors might want to write a more modest conclusion as the severity ("significant") of MR and TR was no part of the analyzes.
Minor comments:
- The title does not give an overview of the content of the manuscript. Please provide a more specified title (e.g. Prognosis of concomitant MR/TR in TAVR)
- What was the median/mean follow up duration?
- Page 11: "However, no studies so far have assessed baseline TR in multivariable analysis of cardiovascular mortality in patients who underwent TAVI. The current study could contribute to filling the gap in the evidence."
As mentioned above, this statement ist not fully correct. McCarty et al. also has performed analysis of TR in TAVR.
Author Response
In the submitted manuscript, Zhang et al. and his Co-authors present data from a large nationalwide database of French which includes al TAVR-patients with concomitant MR/TR. Adjusted analysis showed that the presence of baseline MR and TR had a signifcant impact on patient´s symptoms but were not independent predictors of all-cause mortality.
The authors are to be congratulated on their manuscript, yet their are several aspects, which need to be adressed before the results can be properly interpreted:
Major comments:
- It is indeed a major limitation, that the severity of regurgitation cannot be provided, as relevant MR and TR has been shown to have both independently an effect on the outcome of TAVR patients. Please discuss this issue in more detail in the discussion section.
Mauri V, Körber MI, Kuhn E, Schmidt T, Frerker C, Wahlers T, Rudolph TK, Baldus S, Adam M, Ten Freyhaus H. Prognosis of persistent mitral regurgitation in patients undergoing transcatheter aortic valve replacement. Clin Res Cardiol. 2020 Oct;109(10):1261-1270. doi: 10.1007/s00392-020-01618-9. Epub 2020 Feb 18. PMID: 32072263; PMCID: PMC7515951.
McCarthy, Fenton H.; Vemulapalli, Sreekanth; Li, Zhuokai; Thourani, Vinod; Matsouaka, Roland A.; Desai, Nimesh D.; Kirtane, Ajay; Anwaruddin, Saif; Williams, Matthew L.; Giri, Jay; Vallabhajosyula, Prashanth; Li, Robert H.; Herrmann, Howard C.; Bavaria, Joseph E.; Szeto, Wilson Y. (2018). The Association of Tricuspid Regurgitation With Transcatheter Aortic Valve Replacement Outcomes: A Report From The Society of Thoracic Surgeons/American College of Cardiology Transcatheter Valve Therapy Registry. The Annals of Thoracic Surgery, (), S0003497517315187–. doi:10.1016/j.athoracsur.2017.11.018
Response: Many thanks for your suggestions! We have added the following content quoting the above-mentioned references to the Limitations in the Discussion section.
Increased severity of baseline mitral and tricuspid regurgitation is associated with stepwise increase in short-term comorbidities and worse long-term survival, as well as more rehospitalization for HF in TAVI recipients (Mauri 2020, McCarthy 2018). The lack of information in the specific grading of atrioventricular regurgitation limits the potential extrapolation of the findings from current study into pre-operative risk assessment of TAVI candidates, given the significant different prognostic profile between those with insignificant (none or mild) and significant (moderate or severe) MR or TR(Mauri 2020, McCarthy 2018). Meanwhile, the regression of significant baseline MR and TR was observed in approximately 50-60% of TAVI patients following the procedure, and the post-procedurally regression in regurgitation was significantly associated with better clinical outcomes(Mauri 2020, Yoshida 2019). The unavailability of the dynamic changes of atrioventricular regurgitation over the time course in our study limits further in-depth analysis of the prognostic risk factors in TAVI patients.
- Page 11: "Our results also suggest that physicians in charge of patients with symptomatic AS and MR at a nationwide level properly select the indication for TAVI (instead of possibly an aortic valve surgery with an associated intervention on the mitral valve) since the mortality outcomes after TAVI in the multivariable analysis were not significantly affected by a baseline MR or TR."
This statement is a bit too strong as most of the patients were not sent to surgery because of the high perioperative risk, as you mentioned in the introduction. Current studies (MITAVI; NCT04009434) are investigating whether patients who have had successful transfemoral transcatheter aortic valve implantation (TAVI) with simultaneous moderate to severe mitral valve regurgitation (MR) benefit from additional treatment for mitral valve regurgitation. Please add and discuss this point in the discussion section.
Response: Many thanks for your comment. We have deleted the expression and added the content to the Discussion as follows:
Due to the observations of significant proportion of baseline MR regressed following TAVI procedure, as well as the lack of convincing evidence over the prognostic benefit of simultaneous TAVI and mitral valve clipping for this group of patients with multiple comorbidities and marked risk of adverse events, the intervention on significant MR alongside TAVI is not routinely recommended (Rudolph 2013). A sequential treatment of significant MR following successful TAVI could be potentially beneficial for those who remains symptomatic following the procedure(Stahli 2018). An ongoing clinical trial (MITAVI, NCT04009434) which aims to assess the efficacy of additional mitral valve clipping in TAVI patients with significant MR could potentially fill in the gap of evidence(Linke 2020).
- Conclusion section: The authors might want to write a more modest conclusion as the severity ("significant") of MR and TR was no part of the analyzes.
Response: Many thanks for your comment. We have amended the conclusion as below:
Baseline MR was associated with increased all-cause mortality, cardiovascular mortality, and rehospitalization post-TAVI, but was not associated with a higher rate of stroke. Neither baseline MR nor TR was an independent predictor for all-cause mortality or cardiovascular mortality. Baseline MR was an independent predictor for rehospitalization for HF post-TAVI.
Minor comments:
- The title does not give an overview of the content of the manuscript. Please provide a more specified title (e.g. Prognosis of concomitant MR/TR in TAVR)
Response: We appreciate your suggestion. The title has been changed to: The prognosis of baseline mitral regurgitation in patients with transcatheter aortic valve implantations. We have also amended the running title as: Prognosis of MR in TAVI patients.
- What was the median/mean follow up duration?
Response: Many thanks for your comment. Apologies we didn’t provide the information in the abstract due to the restriction on its wordcount. The follow up duration could be found in the first paragraph of the Results section as follows:
The mean follow-up for included patients was 1.28±1.58 years (median: 0.63 years, interquartile range: 0.03-2.05
- Page 11: "However, no studies so far have assessed baseline TR in multivariable analysis of cardiovascular mortality in patients who underwent TAVI. The current study could contribute to filling the gap in the evidence."
As mentioned above, this statement ist not fully correct. McCarty et al. also has performed analysis of TR in TAVR.
Response: Many thanks for your comment! The study by McCarthy et al. 2018 as kindly mentioned by our reviewer has presented clinical outcomes including all-cause mortality and heart failure readmission in TAVI patients with baseline TR, however, we didn’t manage to find any publications which specifically analyzed the association between baseline TR and cardiovascular-specific mortality in the TAVI population in multivariate Cox regression tests, based on our rather extensive literature search. We have thereby amended the expression as follows:
However, limited studies so far have assessed the association between baseline TR and cardiovascular mortality in multivariate analysis in TAVI recipients. The current study could potentially provide some evidence in the less-studied area.
Round 2
Reviewer 2 Report
I congratulate the authors for their vastly improved manuscript which further nourishes our understanding of concomitant valve diseases in transcatheter aortic valve replacement.